# Necessary and Sufficient Conditions for Compositional Representations

## Abstract

Humans leverage compositionality for flexible and efficient learning, but current machine learning algorithms lack such ability. Despite many efforts in specific cases, there is still absence of theories and tools to study it systematically. In this paper, we leverage group theory to mathematically prove necessary and sufficient conditions for two fundamental questions of compositional representations. (1) What are the properties for a set of components to be expressed compositionally. (2) What are the properties for mappings between compositional and entangled representations. We provide examples to better understand the conditions and how to apply them. E.g., we use the theory to give a new explanation of why attention mechanism helps compositionality. We hope this work will help to advance understanding of compositionality and improvement of artificial intelligence towards human level.

## 1 Introduction

Humans recognize the world and create imaginations in a supple way by leveraging *systematic compositionality* to achieve compositional generalization, the algebraic capacity to understand and produce large amount of novel combinations from known components (Chomsky, 1957; Montague, 1970). This is a key element of human intelligence (Minsky, 1986; Lake et al., 2017), and we hope to equip machines with such ability.

Conventional machine learning has been mainly developed with an assumption that training and test distributions are identical. Compositional generalization, however, is a type of out-of-distribution generalization (Bengio, 2017) which has different training and test distributions. In compositional generalization, a sample is a combination of several components. For example, an image object may have two factor components of color and rotation. In language, a sentence is composed of the lexical meanings and the grammatical structure. The generalization is enabled by recombining seen components for an unseen combination during inference.

One approach for compositional generalization is to learn compositional representations[1], or disentangled representation (Bengio, 2013), which contain several component representations. Each of them depends only on the corresponding underlying factor, and does not change when other factors change. Please see Section 3 for details. Multiple methods have been proposed to learn compositional representations. However, little discussion has been made for some fundamental questions. What kind of factor combinations can be expressed in compositional representation? Though there are some common factor components such as colors and size, what property enable them? When a set of components satisfy the conditions, what kind of mappings are available between the entangled and compositional representations? Can we use the conditions to explain compositionality in conventional models such as attention?

In this paper, we mathematically prove two propositions (Proposition 1.1 and Proposition 1.2) for necessary and sufficient conditions regarding compositional representations. We construct groups for changes on representations, and relate compositional representation with group direct product, and compositional mapping with group action equivalence (Higgins et al., 2018). Then, we use theorems and propositions in group theory to prove the conditions.

---

[1]The word "representation" in this paper refers to variables, not group representation.

**Proposition 1.1** (Compositional representation). A set of components can be expressed compositionally *if and only if* the subgroup product equals to the original group, each component subgroup is normal subgroup of the original group, and the group elements intersect only at identity element.

**Proposition 1.2** (Compositional mapping). Given compositional representation, a mapping is compositional *if and only if* each component has equivalent action in compositional and entangled representations, and for each element of the entangled representation, the orbits intersect only at the element.

Please see Proposition 4.2 and Proposition 4.10 for symbolic statements. We also provide examples to better understand the conditions and how to use them (Section 5). For representations, we see that whether the components can be expressed with compositional representation does not depend only on each component itself, but also on their combination, and the possible values to take. We use the condition for compositional mapping to explain some existing neural network models and tasks, e.g., attention mechanism, spacial transformer and grammar tree nodes. We hope, with these examples, the conditions will be used for validating different compositional representations and mappings, and guiding designs of tasks and algorithms with compositionality. Our contributions can be summarized as follows.

- We propose and prove necessary and sufficient conditions for compositional representation and compositional mapping.

- We provide examples to understand and use the conditions, such as new explanation of attention models.

## 2 RELATED WORK

Human-level compositional learning (Marcus, 2003; Lake & Baroni, 2018) has been an important open challenge (Yang et al., 2019; Keysers et al., 2020). There are recent progress on measuring compositionality (Andreas, 2019; Lake & Baroni, 2018; Keysers et al., 2020) and learning language compositionality for compositional generalization (Lake, 2019; Russin et al., 2019; Li et al., 2019; Gordon et al., 2020; Liu et al., 2020) and continual learning (Jin et al., 2020; Li et al., 2020).

Another line of related but different work is statistically and marginally independent disentangled representation learning (Burgess et al., 2018; Locatello et al., 2019). This setting assumes marginal independence between underlying factors hence does not have compositional generalization problem. On the other hand, compositional factors may not be marginally independent.

Understanding of compositionality has been discussed over time. Some discussions following Montague (1970) uses homomorphism to define composition operation between representations. Recently, Higgins et al. (2018) proposes definition of disentangled representation with group theory. The definition is the base of this paper, and we focus on proving the conditions. Li et al. (2019) defines compositionality probabilistically without discussing conditions to achieve it. Gordon et al. (2020) finds compositionality in SCAN task can be expressed as permutation group action equivalence. This equivalent action is on a component subgroup, but it does not discuss equivalent action on the whole group and the relations between them. There are also other works related to group theory in machine learning (Kondor, 2008; Cohen & Welling, 2016; Ravanbakhsh et al., 2017; Kondor & Trivedi, 2018). However, the previous works do not prove conditions for compositional representation or mapping.

In this paper, we provide and theoretically prove necessary and sufficient conditions for compositional representations and compositional mappings. We use definitions, propositions and theorems from group theory. Please refer to Appendix A. Some of them are summarized in books, such as Dummit & Foote (2004) and Gallian (2012), and we refer to them in the later sections.

## 3 REPRESENTATIONS

In this section, we introduce the definitions of representation and compositional representation used in this paper.

*Representation* in this paper is consistent with the concept in neural network literature. It is a variable, and its value depends on each sample. For example, it can be activations for a layer in a neural network. The values of the activations depend on the network input. Network input and output are also called representations.

*Compositional representation* in this paper means a representation with several separated component representations. It is also called disentangled representation in some literature. "Separated" means that the representation is the concatenation of the component representations. Each component representation corresponds to a underlying component, or a generative factor. When a representation is not compositional, it is an entangled representation.

In the examples in Figure 1, the components are color and shape. The upper images are entangled representations, where color and shape are in the same image. However it is not a compositional representation, because an image is not a concatenation of a color part and a rotation part. The lower vectors are compositional representations, where the left vector is for color and the right vector is for shape.

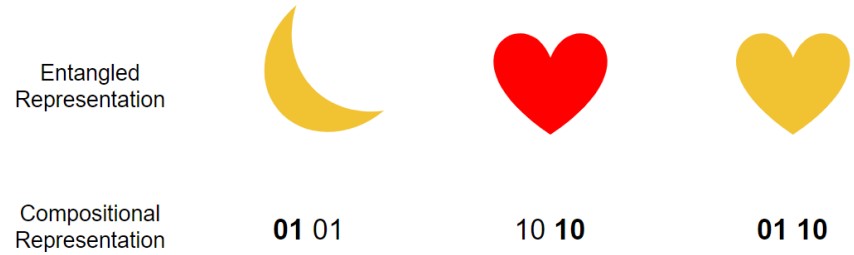

Figure 1: Compositional representations. There are two components of color and shape in this example. A compositional representation can be separated to two component representations (below), but disentangled representations cannot (above).

## 4 NECESSARY AND SUFFICIENT CONDITIONS

In this section, we derive necessary and sufficient conditions for compositionality step by step. We first construct groups for representations. We then describe compositionality with group properties, and study the conditions for them. Based on that, we further study the conditions for mappings between two representations.

### 4.1 GROUPS ON REPRESENTATIONS

Compositionality arises when we compare different samples, where some components are the same but others are not. This means compositionality is related to the changes between samples. These changes can be regarded as mappings, and since the changes are invertible, the mappings are bijective. To study compositionality we consider a set of all bijections from a set of possible representation values to the set itself, and construct a group with the following Proposition 4.1.

**Proposition 4.1.** Let $X$ be any nonempty set and $S_X$ be the set of all bijections from X to itself. The set $S_X$ is a group under function composition[2]. Dummit & Foote (2004) P.29

Since $S_X$ contains all bijections, the group $S_X$ acts on the set $X$ (Definition A.9), and the action is transitive (Definition A.12). We consider two representations and corresponding sets. $X$ is original entangled representation, and $Y$ is compositional representation. We create group $G$ on set $X$, and group $H$ on set $Y$.

### 4.2 COMPOSITIONAL REPRESENTATION

When multiple hidden variables live in the same representation, and cannot be separated by simply splitting the representation, then these variables are entangled in the representation. For example,

---

[2]Function composition is different from compositionality being discussed.

rotation and color are two hidden variables and they are both in a representation of image. We hope to extract the hidden variables by disentangling the representation. Suppose $X$ is a set of entangled representations, and $Y$ is a set of compositional representations. $Y$ has Cartesian product of $K$ small sets $Y_1, \ldots, Y_K$. We hope to find the conditions the changes on $X$ can be expressed by the changes on the components in $Y$.

A component corresponds to a set. For example, color component can take blue, green, etc., from a set of colors. With Proposition 4.1, we can construct a group for each component. With Definition A.2, each of these groups is a subgroup of the original group.

We consider $K$ subgroups. We hope the changes on the entangled representation $X$ are equally expressed by changes on the compositional representation $Y$. This means group $G$ should be isomorphic with the external direct product (Proposition A.1) of the subgroups $H = N_1 \times \cdots \times N_K$. The following Proposition 4.2 has the necessary and sufficient conditions.

**Proposition 4.2.** $N_1, \ldots, N_K$ are subgroups of group $G$. $G$ is isomorphic to the external direct product of the subgroups *if and only if* $G$ is internal direct product of the subgroups. From Definition A.8, we have the following.

$$
G \cong N_1 \times \cdots \times N_K \iff \begin{cases} G = N_1 N_2 \ldots N_K & \text{(A1)} \\ N_i \triangleleft G, \ \forall i = 1, \ldots, K & \text{(A2)} \\ (N_1 \ldots N_i) \cap N_{i+1} = \{e\}, \ \forall i = 1, \ldots, K-1 & \text{(A3)} \end{cases}
$$

*Proof.* " $\impliedby$ ": Theorem A.2. " $\implies$ ": $G$ and $N_1 \times \cdots \times N_K$ are isomorphism, and $N_1 \times \cdots \times N_K$ satisfies the conditions by construction in definition. $\square$

(A1) means the subgroup product should cover the original group. (A2) means all the component subgroups are normal subgroups of the original group. (A3) means the intersection of a subgroup and the previous subgroups only contain the identity element. This corresponds to Proposition 1.1. We will provide examples and look into more details in discussion section.

### 4.3 COMPOSITIONAL MAPPING

We consider to create a mapping between the representations $X$ and $Y$, which we can use to design models. We first consider what property the mapping should satisfy. We then explore conditions for the properties, based on the compositional representation conditions mentioned above. To make the ideas clear, we summarize the relations between sets and groups in Figure 2 (left). We have a isomorphism $\mu$ between group $G$ and group $H$. $G$ is constructed from set $X$ and $H$ is constructed from set $Y$. $\varphi$ is a mapping between $X$ and $Y$. We denote $N_i' = \{h \in H | h_i \in N_i, h_j = e, \forall j \neq i\}$. Then the relations between subsets and subgroups are in Figure 2 (right).

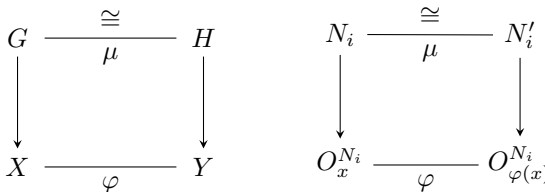

Figure 2: Equivalent group action. We break down conditions for equivalence action on whole representations (left) to each component representation (right) and their relations.

We first consider what property we hope the mapping $\varphi$ to have. We hope both representations always change together. However, the actions are defined for different groups $G$ and $H$. With Proposition 4.5 below, we define action of the same group on both representations.

**Proposition 4.3.** For any group $G$ and any nonempty set $X$ there is a bijection between the actions of $G$ on $X$ and the homomorphisms of $G$ into $S_X$. Dummit & Foote (2004) p.43,p.114

**Lemma 4.4.** The function composition of homomorphisms is a homomorphism. Clark (1984) p.45.

**Proposition 4.5.** For groups $G$ and $H$, $G \cong H$ with bijection $\mu$, $H$ acts on $X$ with homomorphism $\sigma : H \to S_X$, then $G$ acts on $X$ with homomorphism $\sigma \circ \mu : G \to S_X$.

*Proof.* With Proposition 4.3, we only need to prove $\sigma \circ \mu$ is homomorphism. This is true because $\sigma$ and $\mu$ are both homomorphisms (Lemma 4.4). $\qquad\square$

With such action, we look into more details of the requirements. When an action changes one representation, it should always changes the other representation uniquely, which means the mapping should be bijective. Also, the mapping $\varphi$ should preserve the group action. This means the group actions on $X$ and $Y$ are equivalent. (Definition A.13). Note that mapping direction can be either way because the mapping is bijective.

We then consider how to make the action equivalent, with the conditions for compositional representations. We observe that $H$ is external group acting on Cartesian set $Y$, the action of $H$ on $Y$ is product action (Definition A.10). We look at related properties as follows.

**Proposition 4.6.** $N_1 \times \cdots \times N_K$ has production action on $X = X_1 \times \cdots \times X_K$, then $\forall x \in X, \forall i = 1, \ldots, K - 1 : O_x^{N_1 \ldots N_i} \cap O_x^{N_{i+1}} = \{x\}$.

*Proof.* A direct product is isomorphic to itself, so properties in Proposition 4.2 can be used.

$$\forall x \in X, \forall i = 1, \ldots, K - 1, \forall n \in N_1 N_2 \ldots N_{i+1}, n(x) \in O_x^{N_1 \ldots N_i} \cap O_x^{N_{i+1}}$$
$$\implies n \in N_1 \ldots N_i \cap N_{i+1} \underset{\text{Prop 4.2 (C)}}{=} \{e\} \implies n = e \implies n(x) = x$$
$$\implies O_x^{N_1 \ldots N_i} \cap O_x^{N_{i+1}} = \{x\}$$

$\qquad\square$

**Proposition 4.7.** $N_1, \ldots, N_K$ are subgroups of a group acting on a set $X$. If $\forall x \in X, \forall i = 1, \ldots, K - 1 : O_x^{N_1 \ldots N_i} \cap O_x^{N_{i+1}} = \{x\}$, then $\forall x \in X, \forall n_1 \in N_1, n_2 \in N_2, \ldots, n_K \in N_K$,

$$n_1 n_2 \ldots n_K(x) = x \iff n_i(x) = x, \forall i = 1, \ldots, K$$

*Proof.* "$\Longleftarrow$": Repeat for $i = K, \ldots, 1$: $n_1 n_2 \ldots n_K(x) = n_1 n_2 \ldots n_{K-1}(x) = \cdots = x$
"$\Longrightarrow$": $n_1 n_2 \ldots n_K(x) = x \implies n_K(x) = (n_1 n_2 \ldots n_{K-1})^{-1}(x)$
$\implies n_K(x) \in O_x^{N_1 \ldots N_{K-1}} \cap O_x^{N_K} = \{x\} \implies n_K(x) = x$ and $n_1 n_2 \ldots n_{K-1}(x) = x$
Repeat for $i = K - 1, \ldots, 2$, we have $n_i(x) = x, \forall i = 1, \ldots, K$ $\qquad\square$

**Proposition 4.8.** $N_1 \times \cdots \times N_K$ has production action on $X = X_1 \times \cdots \times X_K$, then $\forall x \in X, \forall n_1 \in N_1, n_2 \in N_2, \ldots, n_K \in N_K$, we have $n_1 n_2 \ldots n_K(x) = x \iff n_i(x) = x, \forall i = 1, \ldots, K$.

*Proof.* Proposition 4.6 and Proposition 4.7. $\qquad\square$

Since $H$ and $Y$ are composed by multiple components, we hope to explore whether the equivalence action property can be broken down to conditions on each component, and the relation between components. A natural condition on each component is that the action is equivalent for the component. On $Y$, by its structure, the orbits of each component group action on a element $y$ intersects only at $y$, so we hope this condition also applies to $X$. With Proposition 4.9, we prove in the following Proposition 4.10 that the two conditions together is actually the necessary and sufficient condition for the equivalent action.

**Proposition 4.9.** If $\forall x \in X, \forall i = 1, \ldots, K - 1 : O_x^{N_1 \ldots N_i} \cap O_x^{N_{i+1}} = \{x\}$. $\forall i, \forall x \in X : N_i$ eq. acts on $O_x^{N_i}$ and $O_{\varphi(x)}^{N_i}$, then $\varphi : X \to Y$ is one-to-one.

*Proof.* Any component has bijective mapping, and preserves action. We prove by contradiction.

$$\forall x, x'(x \neq x') \in X, \exists g = n_1 \ldots n_K \in N_1 N_2 \ldots N_K : x' = g(x) \text{ (transitive, Section 4.1)}$$

$$\text{suppose } \varphi(x) = \varphi(x'), \text{ then } \varphi(x) = \varphi(g(x)) = \varphi(n_1 \ldots n_K(x)) \underset{\text{presv.}}{=} n_1 \ldots n_K(\varphi(x))$$

$$\underset{\text{Prop 4.7}}{\Longrightarrow} \varphi(x) = n_i(\varphi(x)) \underset{\text{presv.}}{=} \varphi(n_i(x)), \forall i = 1, \ldots, K \underset{\text{bij.}}{\Longrightarrow} x = n_i(x), \forall i = 1, \ldots, K$$

$$\underset{\text{Prop 4.7}}{\Longrightarrow} x = n_1 \ldots n_K(x) = g(x) = x' \text{ (contradiction)} \Longrightarrow \varphi(x) \neq \varphi(x')$$

$\square$

**Proposition 4.10.** $G \cong N_1 \times \cdots \times N_K$. With $\varphi : X \to Y$,

$$G \text{ eq. acts on } X \text{ and } Y \iff \begin{cases} \forall x \in X, \forall i = 1, \ldots, K-1 : O_x^{N_1 \ldots N_i} \cap O_x^{N_{i+1}} = \{x\} & \text{(B1)} \\ \forall i, \forall x \in X : N_i \text{ eq. acts on } O_x^{N_i} \text{ and } O_{\varphi(x)}^{N_i} & \text{(B2)} \end{cases}$$

*Proof.* " $\impliedby$ ". From Definition A.13, an equivalent action is bijective and preserves action. We first prove the mapping preserves action.

$$\forall g = n_1 n_2 \ldots n_K \in G, \forall x \in X, \varphi(g(x)) = \varphi(n_1 n_2 \ldots n_K(x)) = n_1 n_2 \ldots n_K(\varphi(x)) = g(\varphi(x))$$

We then prove that $\varphi$ is bijection on $X \to Y$. We prove it is one-to-one, onto, well-defined. One-to-one: Proposition 4.9. Onto: $\forall y \in Y, \forall x \in X \exists n_1 n_2 \ldots n_K$ (transitive, Section 4.1) : $y = n_1 n_2 \ldots n_K(\varphi(x)) = \varphi(n_1 n_2 \ldots n_K(x))$, so $\exists x' = n_1 n_2 \ldots n_K(x) \in X : y = \varphi(x')$. Well-defined: $H$ has production action on $Y$, and $\varphi$ is onto, and with Proposition 4.6 and Proposition 4.9, $\varphi^{-1} : Y \to X$ is one-to-one, so $\varphi$ is well-defined.

" $\implies$ ". We first prove (B2). Since subgroup has the same operation with the original group, the equivalent action holds for each component. We then prove (B1).

$$\forall x \in X, \forall i = 1, \ldots, K-1, \forall x' \in O_x^{N_1 \ldots N_i} \cap O_x^{N_{i+1}}, \exists n \in N_1 \ldots N_i, n' \in N_{i+1} :$$

$$x' = n(x) = n'(x) \underset{\text{bij.}}{\implies} \varphi(nx) = \varphi(n'x) \underset{\text{presv.}}{\implies} n\varphi(x) = n'\varphi(x) \implies n^{-1}n'\varphi(x) = \varphi(x)$$

$$\underset{\text{Prop 4.8}}{\implies} n'\varphi(x) = \varphi(x) \underset{\text{presv.}}{\implies} \varphi(n'x) = \varphi(x) \underset{\text{bij.}}{\implies} n'(x) = x \implies x' = x$$

$\square$

To summarize, this proposition says that to find whether a mapping has equivalent action on both representations, we only need to examine whether, for each element, the action is equivalent for each subgroup, and the intersections of orbits only contains the element. This corresponds to Proposition 1.2. In cases both representations are entangled, and we hope to have a compositional representation to connect them, we can use the conditions twice for the mapping. We will discuss the relation to machine learning and compositional generalization, and provide examples in the discussion section.

## 5 DISCUSSIONS

In this section, we provide examples for the conditions of compositional representation and mapping, and look into more insights for better understanding. These examples also serves as applications of the derived results, and we will also discuss about what we learnt from them.

### 5.1 COMPOSITIONAL REPRESENTATION

We provide examples for the boundaries of compositional representation conditions. Proposition 5.1 is used to test normal subgroups.

**Proposition 5.1** (Normal Subgroup Test). A subgroup $H$ of $G$ is normal in $G$ if and only if $xHx^{-1} \subseteq H, \forall x \in G$. Dummit & Foote (2004) p.82 Theorem 6(5), Gallian (2012) p.186 Theorem 9.1.

**Object transformation** We think about examples violating conditions in Proposition 4.2. For a two dimensional geometric object (e.g. image of letter "P"), we consider rotation group $N_1$, and mirror reflection group $N_2$.

If $G = N_1$, and $N_2$ contains non-identity element, then $G \neq N_1 N_2$, because any combination of rotation does not generate a reflection. This violates (A1).

We set $G = N_1 N_2$, and both rotation and reflection take all possible values. In this case, both $N_1$ and $N_2$ are normal subgroups of $G$, $N_1, N_2 \triangleleft G$. However the intersection of $N_1$ and $N_2$ does not only contain identity, $N_1 \cap N_2 \neq \{e\}$. For example, rotating by $\pi$ is equivalent with vertical reflection then horizontal reflection. Therefore, this violates (A3).

If we constrain reflection action to horizontal reflection, and leave rotation to have all possible values, then rotation $N_1$ and horizontal reflection $N_2$ form a group $G$. In this case, $N_1$ and $N_2$ only intersects at identity, $N_1 \cap N_2 = \{e\}$. However, $N_2$ is not normal subgroup of $G$. If we set rotation action $n_1$ to be rotate by $\pi/2$, $n_1^{-1}$ is to rotate by $-\pi/2$. Action of horizontal reflection has $n_2 = n_2^{-1}$. We find $n_1 n_2 n_1^{-1} \notin N_2$, because rotate by $\pi/2$, flip horizontally, and rotate $-\pi/2$, then it does not recover the original by a horizontal reflection. With Proposition 5.1, $N_2$ is not a normal subgroup of $G$, so this violates (A2).

We further think about an example, where we also constrain the rotation to be rotate opposite (by $\pi$), and reflection remains only horizontal. they form group $G$. In this case, rotation and reflection are both normal subgroups of $G$, $N_1, N_2 \triangleleft G$. Also, $N_1$ and $N_2$ only have identity in their intersection $N_1 \cap N_2 = \{e\}$. Therefore, $N_1$ and $N_2$ can be expressed by compositional representation.

We have another example with color as $N_1$ and combination of rotation and reflection as $N_2$. They form group $G$. In this case, regardless the elements in the sets, $N_1$ and $N_2$ are always normal subgroups of $G$, and their intersection always only contains identity. So they can be expressed as compositional representation.

From these examples, we see that whether the components can be expressed with compositional representation does not depend only on each component itself, but also on their combination, and the possible values to take. For some combinations, they are not influenced by possible values.

**Grammar tree node** We also look at an example for language grammar tree node. In a grammar tree, each node $G$ has multiple children. We regard each child as a component $N_i, i = 1, \ldots, K$. We then consider whether the components can be expressed by compositional representation.

For context free grammar, $G = N_1 N_2 \ldots N_K$. Each $N_i$ is normal subgroup of $G$, because a change in one children does not affect others. Also, the intersection of them only contains identity, because each sub-tree is separated. Therefore, a tree node with context free grammar is possible to be expressed with compositional representation.

We also look at an example of root node in syntactic tree for fixed length sentences. When at least one subgroup actions change the phrase length, the product of $N_i$ may be not a group, because it may change the sentence length ($G \neq N_1 N_2 \ldots N_K$). This means $G$ does not fit the conditions. Note that such grammar is not a context free grammar.

## 5.2 Compositional mappings

Conditions for compositional mapping can be used to design models for the relation between two representations. We first describe the connection with machine learning, and then use the conditions to explain some existing neural network models and tasks.

**Model training and architecture for compositional generalization** We consider samples in training. For compositional generalization, some elements in the whole set $Y$ (or $X$) are missing, but for each subset $Y_i$, the elements are complete. For condition (B2) in Proposition 4.10, each subset has complete samples, so it is satisfied. For condition (B1), the set has missing elements, so we do not have information to tell whether it is satisfied. To address this problem for condition (B1), we may constrain the mapping $\varphi$ that the images for each component intersect at only one element.

**Attention mechanism**    Attention mechanism is used for compositional modelings (Goyal et al., 2019; Russin et al., 2019; Li et al., 2019). We consider a problem that there are two components for an object. One component is the position of the object, and the other is the local shape (or word for language processing) of the object. We look into an attention network that combines the two components to generate output.

We first check whether the two components can be expressed as compositional representation. Set $N_1$ is group for position, $N_2$ is group for shape, and $G = N_1N_2$. For an object, if we change shape, position, and shape back, the shape does not change. Similarly, change position, shape and position back, the position does not change. With Proposition 5.1, $N_1$ and $N_2$ are both normal subgroups of $G$. Also, $N_1 \cap N_2 = \{e\}$ because changing position does not change shape, and changing shape does not change position. This means the components can be expressed as compositional representation.

We then check whether the model is compositional mapping. First, we look at each component. For both position and shape, the mapping is bijective and preserves action. Second, we look at the orbits of images. Since the shape only changes locally, it does not change position, and position does not change shape. Therefore, the model is compositional.

Note that we do not assume that the attention is sparse for each sample. This is different from some conventional explanations (Bengio, 2017; Goyal et al., 2019) of attention mechanism. The attention helps compositional generalization not because it is dynamically sparse, but it fits the conditions. For example, when there are multiple positions to attend in one attention map, it may still fit the conditions.

**Spatial transformer**    In the example of attention mechanism, position is aligned with one build-in dimension of data structure. However, this is not necessary. Here, we provide another example with Spatial Transformer (Jaderberg et al., 2015) for such a case. We focus on the transformations for rotation and scaling. The data structure does not have such build-in dimensions. We see that rotation and scaling satisfy Proposition 4.2 to be expressed with compositional representation, and the mapping satisfies Proposition 4.10 to be compositional mapping.

However, if we consider rotation and shape, the network might not be compositional. For example, rotations by $0, 2\pi/3, 4\pi/3$ map a triangle it to itself, but this does not apply to a square. This means if a set of rotation contains both 0 and $2\pi/3$, it is not bijective for triangle. If it contains 0 but not $2\pi/3$, it is not bijective for square. Therefore, it violates (B2).

**Ambiguous context free grammar**    We discussed that a node for context free grammar is able to be expressed with compositional representation. However, when there is syntactic ambiguity, we are not able to get the compositional mapping. This violates the condition (B1), because the orbits have more than one elements in the intersection.

## 6  CONCLUSION

We use group theory to prove necessary and sufficient conditions for compositional representation and mapping. We discuss examples for the conditions, and understand the boarders of them. We also provide new explanations for existing methods. We expect the conditions will help validating compositional representations and mappings, and guide designs of tasks and algorithms. We hope this work will help to advance compositionality and artificial intelligence research.

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

# A PRELIMINARIES FOR GROUP THEORY

In this section, we go through some preliminaries for group theory. We provide widely used definitions, propositions and theorems, with references for more details.

## A.1 GROUPS

**Definition A.1** (Group). A group is an ordered pair $(G, \circ)$ where $G$ is a set and $\circ$ is a binary operation on $G$ satisfying associativity, identity $e$ and inverses. We say $G$ is a group if the operation $\circ$ is clear from the context, and we also omit $\circ$. Dummit & Foote (2004) p.16 (also Gallian (2012) p.43)

**Definition A.2** (Subgroup). If a subset $H$ of a group $G$ is itself a group under the operation of $G$, we say that $H$ is a *subgroup* of $G$. Gallian (2012) p.61 (also Dummit & Foote (2004) p.46)

**Definition A.3** (Normal subgroup). A subgroup $H$ of a group $G$ is called a *normal subgroup* of $G$ if $aH = Ha, \forall a \in G$. We denote this by $H \triangleleft G$. Dummit & Foote (2004) p.82, Gallian (2012) p.185

## A.2 MAPPINGS

**Definition A.4** (Group homomorphism). A *homomorphism* $\varphi$ from a group $G$ to a group $H$ is a mapping from $G$ into $H$ that preserves the group operation, i.e., $\forall a, b \in G, \varphi(ab) = \varphi(a)\varphi(b)$. Gallian (2012) p.208, Dummit & Foote (2004) p.36.

**Definition A.5** (Group isomorphism). The map $\varphi : G \to H$ is called an *isomorphism* and $G$ and $H$ are said to be isomorphic or of the same isomorphism type, written $G \cong H$, if $\varphi$ is a homomorphism and a bijection. Dummit & Foote (2004) p.37, Gallian (2012) p.128.

## A.3 PRODUCTS

**Definition A.6** (Product of subgroups). Let $H_1, \ldots, H_K$ be subgroups of a group and define $H_1 H_2 \ldots H_K = \{h_1 h_2 \ldots h_K | h_i \in H_i, \forall i = 1, \ldots, K\}$. Dummit & Foote (2004) p.93

**Definition A.7** (External direct product). Let $G_1, \ldots, G_K$ be a finite collection of groups. The *external direct product* of $G_1, \ldots, G_K$, written as $G_1 \times \cdots \times G_K$, is the set of all $K$-tuples for which the $i$th component is an element of $G_i$ and the operation is component wise. In symbols,

$$G_1 \times \cdots \times G_K = \{(g_1, \ldots, g_K) | g_i \in G_i\}, \quad (g_1, \ldots, g_K)(g_1', \ldots, g_K') = (g_1 g_1', \ldots, g_K g_K')$$

Gallian (2012) p.162, Dummit & Foote (2004) p.152

**Proposition A.1** (External direct product is a group). If $G_1, \ldots, G_K$ are groups, their external direct product is a group. Dummit & Foote (2004) p.153. Proposition 1.

**Definition A.8** (Internal direct product). Let $H_1, \ldots, H_K$ be a finite collection of normal subgroups of $G$. We say that $G$ is the *internal direct product* of $H_1, \ldots, H_K$, if

$$G = H_1 H_2 \ldots H_K \qquad \text{and} \qquad H_1 H_2 \ldots H_i \cap H_{i+1} = \{e\}, \ \forall i = 1, \ldots, n-1$$

Gallian (2012) p.197, Dummit & Foote (2004) p.172.

**Theorem A.2** (Recognition theorem). If a group $G$ is the internal direct product of a finite number of subgroups $H_1, \ldots, H_K$, then $G$ is isomorphic to the external direct product of $H_1, \ldots, H_K$. Gallian (2012) p.198, Dummit & Foote (2004) p.171.

## A.4 GROUP ACTIONS

**Definition A.9** (Group action). A *group action* of a group $G$ on a set $X$ is a map from $G \times X$ to $X$ (written as $g(x), \forall g \in G, x \in X$) satisfying the following properties:

$$g_1(g_2(x)) = (g_1 g_2)(x), \forall g_1, g_2 \in G, x \in X \qquad \text{and} \qquad e(x) = x, \forall x \in X$$

Dummit & Foote (2004) p.112

**Definition A.10** (Product action). $G = G_1 \times \cdots \times G_K$ is a group, and $X = X_1 \times \cdots \times X_K$ is a set. $G$ acts on $X$ by the rule $(g_1, \ldots, g_K)(x_1, \ldots, x_K) = (g_1 x_1, \ldots, g_K x_K)$. Cameron et al. (2008), Praeger & Schneider (2018) p.71.

**Definition A.11** (Orbit). Let $G$ be a group action on the nonempty set $X$. The equivalence class $O_x^G = \{g(x)|g \in G\}$ is called the *orbit* of $G$ containing $x$. Dummit & Foote (2004) p.115

**Definition A.12** (Transitive action). The action of $G$ on $X$ is called *transitive* if there is only one orbit, i.e., given any two elements $x, y \in X$ there is some $g \in G$ such that $x = g(y)$. Dummit & Foote (2004) p.115

**Definition A.13** (Equivalent action). Two actions of a group $G$ on sets $X$ and $Y$ are called *equivalent* if there is a bijection $\varphi : X \to Y$ such that $\varphi(g(x)) = g(\varphi(x))$ for all $g \in G$ and $x \in X$. We say $G$ eq. acts on $X$ and $Y$. Lovett (2015) p.385.

