# OpenReview forum: "Necessary and Sufficient Conditions for Compositional Representations"
_ICLR.cc/2021/Conference — Reject_

### Official Review · AnonReviewer2 · 2020-10-14
**Needs definitional and presentational development**

**Rating:** 3
**Confidence:** 4

**Review:**

Post-revision update
----------------------------

Thanks to the authors for their revision. Unfortunately, I still feel the contribution and value of the work is not well communicated. Many of my previous points still stand. It is not a sufficient response to just say that the two propositions are the main contribution and the critical summary of findings and their relevance.

Original Review
------------------------

This paper presents a group-theoretic approach to compositionality, and attempts to provide necessary and sufficient conditions for the possibility of compositional representations. While this is an admirable goal, I am concerned that the paper does not live up to it. I recommend rejection at present, unless the definitions and presentation are substantially clarified.

* The authors have assumed a particular definition of "compositional" for "compositional representations," but they have neither clearly stated a definition for compositional representations nor justified it. Section 4.2. does not clearly state a definition of compositionality.
    * The authors cite Bengio (2013) as a reference when they introduce the idea of learning compositional representations, but this term is not defined there. In their mathematical definitions, they appear to assume compositional = componential.
    * However, what it means for a representation to be compositional is actually quite difficult to assess, some of the papers that the authors cite (e.g. Andreas, 2019) discuss this. See also Zadrozny (1992).
    * The authors definition of compositionality seems to my view to be closer to a notion of disentanglement. Indeed, their definition of compositionality itself seems to closely reflect the disentanglement definition given in Higgins et al. (2018). The authors note that representations could still be compositional even if the features are not statistically independent, which is valid, but does not contradict the point I am making here.
    * My point is that, from Montague, we know we can define compositionality in terms of any composition operator in representation space. The authors do not justify that *any* composition operator could be specified in terms of their definition of compositionality. In fact, it seems clear that assuming a product of a finite number of subgroups could *not* capture the original linguistic goals of compositionality, which was *precisely* to allow *infinite recombination of elements to arbitrary depth*. For example, the grammar of arithmetic symbols allows arbitrarily long sequences of operations. How could this structure be experessed in this framework?
    * It appears to me tha this article is therefore not about compositionality per se, but rather about disentanglement of a finite number of subgroups underlying the data. This is also a subject worth investigating, but the paper framing should reflect this, and the relation to prior work should clarify more fully how it differs from prior group-theoretic definitions of disentanglement (e.g. Higgins et al., 2018).

* Other definitions are not clearly expressed either. For example, what are the "original" representations that the authors refer to, where do they come from? This entails certain assumptions about e.g. the mapping from possibly noisy data into the representational spaces that are essential to understanding whether these ideas have any meaning in practice.

* In general, the authors focus too much on formal notation, without developing intuitions for what results mean and why they would be useful. The authors present a page of group theory definitions, but this seems almost certain to be unhelpful. Any reader who is not familiar with group theory will not be able to develop sufficient intuitions to understand the rest of the paper from reading these terse definitions without examples, and to those of us familiar with group and category theory this is space that could be much better spent on clarifying and elaborating the exposition of the contributions.

In summary, I think that while this work could present a useful contribution to understanding disentanglement, it would need substantial rethinking of the definitions, more connection with the prior work, and clarification of the writing and exposition for me to see it as ready to present.

References
---------

Zadrozny, Wlodek. "On compositional semantics." COLING 1992 Volume 1: The 15th International Conference on Computational Linguistics. 1992.

(Other references cited above are referenced in the article.)

---

> ### Author Response · Authors · 2020-11-19
> **Reply to Reviewer 2**
>
> Thank you for your questions and suggestions.
>
> Q1: Definition of compositionality and relation to previous work.
>
> A1: Please see the common reply for the definition. Higgins et al. (2018) focuses on proposing compositionality definition. However, this work focuses on the proof of the conditions.
>
> Q2: What compositionality means for a representation to be compositional is actually quite difficult to assess, some of the papers that the authors cite (e.g. Andreas, 2019) discuss this. See also Zadrozny (1992).
>
> A2: In experiments, compositionality is difficult to assess, so that sometimes people measure compositional generalization, which is relatively straight-forward to evaluate. Also, this paper is about theory, and it is less influenced by practical difficulties.
>
> Q3: My point is that, from Montague, we know we can define compositionality in terms of any composition operator in representation space. The authors do not justify that any composition operator could be specified in terms of their definition of compositionality.
>
> A3: The composition operator shall correspond to homomorphism in this paper.
>
> Q4. In fact, it seems clear that assuming a product of a finite number of subgroups could not capture the original linguistic goals of compositionality, which was precisely to allow infinite recombination of elements to arbitrary depth. For example, the grammar of arithmetic symbols allows arbitrarily long sequences of operations. How could this structure be expressed in this framework?
>
> A4: A subgroup can be a product of subgroups. This recursive structure is able to express arbitrary depth for infinitely long sentences. The second proposition is also recursive. However, this might be an extended topic for this paper.
>
> Q5: Other definitions are not clearly expressed either. For example, what are the "original" representations that the authors refer to, where do they come from? This entails certain assumptions about e.g. the mapping from possibly noisy data into the representational spaces that are essential to understanding whether these ideas have any meaning in practice.
>
> A5: The original representation is an entangled representation where different components are not separated. We changed it in the manuscript.
>
>
> Q6: In general, the authors focus too much on formal notation, without developing intuitions for what results mean and why they would be useful. The authors present a page of group theory definitions, but this seems almost certain to be unhelpful.
>
> A6: We like to add more intuitions and move the group theory definitions to appendix. Please find why they are helpful in the application part in the common reply.

---

### Official Review · AnonReviewer4 · 2020-10-20
**Interesting characterization of conditions pertaining to compositionality using group theory. While topic and analysis are timely, ultimately it remains unclear how much light the results sheds on compositionality.**

**Rating:** 4
**Confidence:** 3

**Review:**

##### Summary #####
This work uses group theory to investigate compositionality. It characterizes conditions (1) for a set of components to be composed and (2) for a mapping to be compositional.


##### Reasons for score #####
My vote leans toward rejection. Compositionality has drawn the attention of researchers for many years and the goal to clarify what we mean by it is certainly worth pursuing. The consequence of Proposition 1.1, on the conditions of compositional components, that composition depends not only on each component, but also on their combination, is particularly interesting and highlights how this approach can fruitfully inform our understanding of compositionality. The results also look sound to me. However, I am ultimately not certain about the progress, conceptual or technical, this contribution makes toward our understanding of object of study. In other words, to my mind, there is a lack of motivations, discussion, or applications to contextualize Proposition 1.1 and Proposition 1.2. I see how both Section 1 and Section 5 could serve this purpose but they fall a bit short. I should stress that this project is quite interesting, but it would greatly benefit from working out --or clarifying, in case I missed it-- what we learn from these results.

#### Pros ####
+ Characterizing compositionality is fundamental to our understanding of human cognition. It might, consequently, also be important for the development of machine learning techniques. While, prima facie, compositionality is pretty intuitive, its formal definition has proven to be quite elusive. Efforts to formally characterize it are much needed
+ Proposition 1.1 and Proposition 1.2, the main results of the paper, answer some fundamental questions and have a few interesting consequences.
+ The exposition surrounding the main results (sections 3 and 4) is incremental and well-executed


#### Cons ####
- My main concern is that this work does neither succeed in linking obtained results to established literature and standing discussions nor in highlighting their relevance for researchers working on this topic. This is a major concern since it certainly has a lot of potential. Since this research addresses fundamental questions about the object of study, I would encourage the authors to clarify how the two propositions and their consequences inform our understanding of compositionality (i.e., what do we learn that we didn't before?). The conclusion is a good example of where this paper falls short: There is no critical summary of findings and their relevance, nor discussion of further research. Instead, it is an itemization of what was done.
- The writing is, at times, hard to follow.

---

> ### Author Response · Authors · 2020-11-19
> **Reply to Reviewer 4**
>
> Thank you for comments and suggestions.
>
> Q1: There is a lack of motivations, discussion, or applications to contextualize Proposition 1.1 and Proposition 1.2.
>
> A1: Please see the common reply.
>
> Q2: It would greatly benefit from working out what we learn from these results.
>
> A2: What we learn is the two propositions, and their applications are in A1.
>
> Q3: My main concern is that this work does neither succeed in linking obtained results to established literature and standing discussions nor in highlighting their relevance for researchers working on this topic.
>
> A3: The connection is that the previous works do not prove the conditions, and we do. The relevance is possible applications discussed in A1.
>
> Q4: I would encourage the authors to clarify how the two propositions and their consequences inform our understanding of compositionality (i.e., what do we learn that we didn't before?).
>
> A4: The two propositions themselves are the main of what we learnt. Also, there are some specific findings in discussion sections.
>
> Q5: There is no critical summary of findings and their relevance, nor discussion of further research. Instead, it is an itemization of what was done.
>
> A5: The critical summary of findings are summarized as the two propositions.

---

### Official Review · AnonReviewer1 · 2020-10-28
**Helpful idea for understanding compositionality, but hard to extract the lessons**

**Rating:** 3
**Confidence:** 4

**Review:**

# Overall review

This paper applies concepts from group theory to help find necessary and sufficient representations on the presence of compositionality in representations and on mappings between them.  This topic is very important, as people tend to use "compositional" in many ways, often not explicitly defined.  The paper, however, is hard to follow, because the main concepts and theorems are not adequately illustrated with examples.  In the final section, when examples are provided, it's still not clear how they apply to representation learning, the topic of this conference.  Thus, while the topic is very relevant and the approach welcome, it is hard to know what lesson we have learned from the theorems in the paper.

Pros:
* Seeks to clarify and precisify the definitions of compositional representations, and find necessary and sufficient conditions on their existence.
* Contributes theoretical results on compositionality, a timely topic.

Cons:
* The text is hard to follow without motivating examples of the definitions and theorems.
* The applications in the discussion section are somewhat opaque.  The application to attention doesn't appear to make any explicit reference to the attention mechanism at all.
* While theoretical understanding is extremely valuable, it's unclear how to use the results to detect or promote compositionality as usually understood.


# Minor comments

* "In language, a sentence is a combination of grammar and lexicon."  It might be more informative here to say that the meaning of a whole sentence is composed from the meanings of the parts and the grammatical structure of the sentence.

* At the end of the introduction, the statements of Propositions 1.1 and 1.2 are hard to understand without later background from the paper.  Is it possible to provide a more intuitive statement (even if not fully precise) for these?

* "We will provide examples and look into more details in discussion section."  For this reader at least, it would have helped to have examples more thoroughly laid out in Sections 3 and 4.  For example: what structure of groups is being assumed for X and Y in 4.1-4.2?  Y is assumed to be compositional representations, but we haven't been given a clear definition of that concept or an exact example.


# Typographic comments:

* I think e in definitions 3.8 and 3.9 refers to the identity element of the group, but this should be stated explicitly (e.g. in the definition of a group).

* Defn 3.12: I'd use X instead of A here for consistency.

---

> ### Author Response · Authors · 2020-11-19
> **Reply to Reviewer 1**
>
> Q1: The text is hard to follow without motivating examples of the definitions and theorems.
>
> A1: We are going to add an example.
>
> Q2: The applications in the discussion section are somewhat opaque. The application to attention doesn't appear to make any explicit reference to the attention mechanism at all.
>
> A2: In attention mechanism, query and keys generate attention maps. Attention map and attended values are the two components in the example.
>
> Q3: While theoretical understanding is extremely valuable, it's unclear how to use the results to detect or promote compositionality as usually understood.
>
> A3: Please refer to the application part in common reply.

---

### Official Review · AnonReviewer3 · 2020-10-28
**Ambitious idea, but the presentation is flawed.**

**Rating:** 3
**Confidence:** 4

**Review:**

** Summary **

The paper attempts to formally explore the necessary and sufficient conditions for compositional representations, leveraging the formal tools from group theory. While the ideas look potentially promising, the presentation is fundamentally flawed, with certain key notions left without formal definitions. As a consequence, it becomes impossible to estimate the theoretical significance of the contribution or use the obtained results in practice.

** Strengths **

The paper addresses a highly relevant and important problem. In general, it would be extremely helpful for a broad range of ML and AI researchers and practitioners if we were to formally describe the conditions in which compositional representations can be obtained.

** Weaknesses **

Unfortunately, the proposed approach is not described clearly enough for it to be widely useful.

In general, I believe that when formal tools (like group theory) are applied to prove anything outside of their original domain (i.e. when we are using group theory to reason about compositional representations in machine learning),
it is crucial to 1) clearly define all involved notions (not only mathematical, but also the ones to which mathematical tools are applied) 2) clearly motivate the application.

** Clarity **

Unfortunately, the clarity of the contribution is not up to the standards of ICLR conference. In general, I believe that clarity concerns are secondary to other evaluation components (experimental support, novelty, etc.). In this case, however, it becomes impossible for me to evaluate other components because I can not fully understand the approach from its description.

For example, while the paper is focused on compositional representations, the actual description/definition of what exactly authors mean by compositional representations comes only on the 4th page (after some formal results were already stated).

The description is as follows: "Compositionality arises when we compare different samples, where some components are the same but others are not. This means compositionality is related to the changes between samples. These changes can be regarded as mappings, and since the changes are invertible, the mappings are bijective. To study compositionality we consider a set of all bijections from a set of possible representation values to the set itself, and construct a group with the following Proposition 4.1.". At the same time, there was no formal definition of "representation" before that paragraph. In the very next paragraph, however, the authors say "We consider two representations and corresponding sets. X is original entangled representation, and Y is compositional representation".

The concerns I described above are related to the overall structure of the contribution. A separate and also a major concern is that the writing itself should be improved too. There are numerous confusingly phrased sentences which make reading difficult.

For example, we can take a look at the very first sentence in the abstract: "Humans naturally use compositional representations for flexible recognition and
expression, but current machine learning lacks such ability". It's not clear what is meant by "recognition and expression", it also seems unnatural to say that "machine learning" lacks a certain ability,
because machine learning is a field of study. It may be better to rephrase it to "machine learning methods". While these concerns are minor, they are ubiquitous thoughout the paper, which substantially hinders readability.

** Suggestions **

The direction may be promising, but unfortunately, the paper needs a thorough reorganization in order to be publishable.

I would like to suggest moving the standard group theory definitions from the main text into appendix. Some of the proofs could be moved there too. The space obtained this way may be used to
1) formally define a) what a "representation" is b) what a "compositional representation" is c) the general problem setting
2) motivate the chosen definitions with some potential applications. I realize that the examples in the end of the article are intended to serve that goal, but in my opinion, neither of them is explored in enough detail.

I understand that a lot of work went into this article, and I hope that the authors won't feel discouraged by the feedback, but use it as an opportunity to improve the paper.

** Update after the authors' response **

I have read the authors' response and other reviews. I still believe that my evaluation is correct at the moment.

At the same time, I believe that the research direction is very and promising, and I hope to see the updated version of the manuscript published in the future!

---

> ### Author Response · Authors · 2020-11-19
> **Reply to Reviewer 3**
>
> Thank you for your suggestions. We hope to improve the paper with them. Please also see the common reply.

---

### Author Response · Authors · 2020-11-19
**Common reply**

Thank you for constructive suggestions. Here are answers to some common questions.

Q1: Definitions for representation and compositional representation:

A1: Representation in this paper is consistent with the concept in neural network literature. It is a variable, and its value depends on each sample. For example, it can be a layer in a neural network. The values of the layer depends on the network input. Network input and network output are also called representations.

Compositional representation in this paper means a representation with several separated component representations. It may be also called disentangled representation (the concepts may have somewhat different meanings in different settings). “Separated” means that the representation is the concatenation of the component representations. Each component corresponds to a generative factor, such as color or shape.

For example, an image can be a representation with two components of color and shape. However it is not a compositional representation, because an image is not a concatenation of a color part and a shape part. If we have a color detector and shape detector to extract color variable and shape variable, and concatenate them, then it is a compositional representation.

Q2: Motivating applications

A2: This work has direct applications that may contain two folds. First, whether the compositional representation or mapping exist, and if not, it saves researchers much time and effort. Second, guide and validate algorithm designs.

The condition for compositional representation can be used to detect whether the set of components are valid. It may also be used to design components. The condition for compositional mapping can be used to detect whether such mapping exists, and maybe help to guide designing learning algorithms, and validate them.

For indirect application, compositional representation is helpful in different areas, such as compositional generalization, and previous works have been discussed in more detail.

---

> ### Comment · AnonReviewer2 · 2020-11-19
> **Thanks, I look forward to the revised version, here are some further suggestions for it.**
>
> I'll address both your specific comments and the common response here. Thanks for the replies, I do think they clarify some of the definitions and assumptions you are making, and I appreciate that you will add more definitional clarity and  intuitions to the revised paper. I look forward to reviewing the revised version to see the improvement, and updating my score accordingly. I have some further suggestions that I think would make the revised version much stronger from my perspective.
>
> **Most importantly, I would like to make a more specific request for some intuitions to provide in the paper.** In particular, you highlight two applications:
> 1) To detect whether a compositional representation or mapping exists.
> 2) To guide and validate algorithm designs.
>
> To really motivate that your results are *actually* useful for these applications, I would like an intuitive example for each in which your results make something clear that wasn't clear before. For example, you provide the example of color and object being entangled in an image, but disentangled in a representation, which reflects a common application of disentanglement. Is there something about this example which is uniquely clarified by your results? Or is there another intuitive example where your results would change our perspective on what is compositionally representable and what is not? If you provide such an example and explain clearly how it improves our understanding beyond what would be provided by e.g. the definition of disentanglement/compositionality that you are already assuming, I think it would really help readers appreciate the impact of your work.
>
> Similarly, for the algorithms case, can you give an example of an existing algorithm that makes more or less sense after considering your results, compared to what we would think from the definition of disentanglement alone? Or can you give an example of what the direct implications would be for new algorithms? This would be very helpful for readers to identify when we should consider your results when developing new algorithms.
>
> Importantly, in both cases you should clearly articulate the advantage that your results provide over what we could infer from just the definition of disentanglement/compositionality that you are assuming.
>
> As a secondary point, I feel that the disentanglement vs. compositionality terminology could be discussed further: why are you choosing to use the term compositionality instead of disentanglement, and what are the distinctions you see between the two? Some discussion of this could help place your work in the broader literature better.

---

> > ### Author Response · Authors · 2020-11-25
> > **Thank you for suggestions**
> >
> > Thank you for your suggestions. We uploaded an updated manuscript. The definitions are clarified at Section 3, and the applications are summarized in the last paragraph of Introduction (before contributions).
> >
> > Q: The disentanglement vs. compositionality terminology could be discussed further: why are you choosing to use the term compositionality instead of disentanglement, and what are the distinctions you see between the two?
> >
> > A: We use the term “compositional” because “disentangled” representations are sometimes assumed to have statistically independent components. So we hope to avoid potential misunderstanding as this paper does not assume that. We think otherwise they are the same.

---

### Decision · Program_Chairs · 2021-01-07
**Final Decision**

**Decision:**

Reject

**Comment:**

The reviewers were clearly excited by the novel application of group theory to the problem of composition, and think the core idea is good.  However, the reviewers also expressed concern about the clarity of the paper, stating that in several places examples might help. Reviewers were also interested in seeing the work tied to real world applications, and how the work expands our existing knowledge about the composition of learned representations.  I hope their suggestions will help the authors to turn this into a stronger, clearer paper.